# Cooling Effect of Paddy on Land Surface Temperature in Cold China Based on MODIS Data: A Case Study in Northern Sanjiang Plain

**Guoming Du [1], Wenqi Liu [1], Tao Pan [2,3,4,5,6,*], Haoxuan Yang [1] and Qi Wang [1]**

[1] School of Public Administration and Law, Northeast Agricultural University, Harbin 150030, China; nmgdgm@126.com (G.D.); liuwenqi67@163.com (W.L.); yhx965665334@163.com (H.Y.); Edwangqi@163.com (Q.W.)

[2] State Key Laboratory of Desert and Oasis Ecology, Xinjiang Institute of Ecology and Geography, Chinese Academy of Sciences, Urumqi 830011, China

[3] Department of Geography, Ghent University, 9000 Ghent, Belgium

[4] Sino-Belgian Joint Laboratory of Geo-information, Xinjiang Institute of Ecology and Geography, Chinese Academy of Sciences, Urumqi 830011, China

[5] Sino-Belgian Joint Laboratory of Geo-information, Ghent University, 9000 Ghent, Belgium

[6] University of Chinese Academy of Sciences, Beijing 100049, China

* Correspondence: tao.pan@ugent.be

**Abstract:** Fast-growing crops have been evolved in North China, accompanied by intense paddy expansion, leading to dramatic impacts on the agricultural environment. Among these environmental issues, the impact of paddy expansion on land surface temperature is still unclear. In the present study, based on Landsat images and MODIS land surface temperature (LST) products, the crop pattern and monthly LST in the northern Sanjiang Plain are obtained. A 1 km scale grid unit is built to investigate the relationship between LST and paddy expansion. The results obtained from the study are as follows. Firstly, for crop patterns, cropland planting is given priority to paddy fields, accompanied by an aggregated pattern, while upland crops present a discrete pattern. Secondly, for LST changes during the growing season, the maximum LST occurs in June, and the lowest values occur in October across the whole region. In addition, the LST of paddy fields is lower compared with that of upland crops for the whole growing season. Thirdly, at the 1 km grid scale, the relationship between monthly LST and paddy field ratio is significantly negative, and better represented by a cubic function rather than a linear fit. Finally, LST decreases with the increased fraction of the rice paddy area more rapidly when rice paddy is aggregated and accounted for by more than 80% of each study grid. The findings of this study are important to guide agricultural production and to better understand the environmental effects of paddy expansion in cold regions.

**Keywords:** regional climate change; paddy expansion; land surface temperature (LST); remote sensing; correlation and regression analysis; agriculture

## 1. Introduction

The "scientific planning and implementation strategy" proposed by the Global Land Programme (2016–2021) states that the land system is the place where the interactions between human beings and natural environments occur, and that the land system evolution has an important impact on the socio-economic system. As a vital branch of the land system, cropland spatial distribution evolution will inevitably lead to changes in ecological conditions. Previous literature has indicated that great differences in surface thermal environments have existed between different underlying

land surface types, and the effect of underlying land surface changes on the land surface energy balance is significant [1–5]. This issue was more obvious in farmland regions [6]. Under different planting patterns, the apparent coverage conditions of cropland in crop growth seasons were different because of different farming patterns. For example, water injection was required in the early stages of the planting process in paddies, while it was not required in upland crops [7–9]. The existence of underlying land surface water and its quantity directly affects changes in environmental factors such as soil moisture [10]. This has led to obvious differences in regional land surface temperature between different crop planting types at different times. A number of studies have investigated the effects of croplands on the surrounding environment. For example, irrigated agriculture, moderated daily maximum air temperature, and increased daily minimum air temperature narrow the range of diurnal change in air temperature [11]. The extent of the effects of rice fields on air temperature and humidity can vary depending on growth stages and depending on cultivars with different physiologies [12,13]. Furthermore, investigation of the effect of rainfall on methane release during the paddy water release process revealed that greenhouse gas emissions during paddy field planting affected the environment of surrounding land [14–16]. Moreover, the effect of different rice varieties on methane release during the cultivation of tropical paddies also affected the local thermal environment [17,18]. Most of the previous literature has been dominated by the climate effect of rice in terms of its contribution to air temperature and humidity, as well as greenhouse gas release [19–21]. However, from the perspective of underlying land surface differences in cropland, studies regarding the effect of paddy expansion on land surface temperature have rarely been reported.

The Sanjiang Plain is a major grain production area and also a commercial grain base in China [22,23]. Global warming has provided an external environment for paddy field expansion in this cold region since the beginning of the 21st century, particularly via a northward shift of the rice planting boundary [24,25]. Previous literature has indicated that paddy fields in the Sanjiang Plain increased by 60% from 2000 ($4.77 \times 10^4$ km$^2$) to 2014 ($7.63 \times 10^4$ km$^2$) [26]. From 2010 to 2015, the paddy area created from the conversion of upland crops in Northeast China accounted for 91.7% of the total conversion areas of the whole of China [25]. It is noteworthy that the expanding water body and changing vegetation coverage during the crop growth stage of paddy fields affects land surface thermal environments [27–29]. The premise of an unrestricted expansion of paddy fields may change the steady-state of the land surface thermal environment. As an indispensable factor of crop growth, changes in land surface temperature may affect the growth of crops and total grain production [30].

The present study investigates the relationship between land surface temperature and paddy expansion, derived from remotely sensed products, including Landsat OLI and MODIS LST. The objectives of this study are as follows: (1) to explore the spatial distribution of crop patterns, including paddy fields and upland crops; (2) to compare the land surface temperature differences among paddy fields and upland crops during the whole growing season; (3) to investigate the relationships between paddy fields and land surface temperatures for different paddy field ratios; and (4) to quantify the threshold of the cold island effect caused by paddy expansion.

## 2. Materials and Methods

### 2.1. Study Area

"Cold China" refers to the northernmost province of China (43°26′–53°33′ N and 121°11′–135°05′ E). The region is located in a low temperature zone, with annual average temperature lower than 1 °C [31]. The study area covered the north of Sanjiang Plain, which is located in the east of cold China, spanning the borders of China and Russia between 45°43′–48°24′ N and 129°10′–135°3′ E. The northern Sanjiang Plain belongs to the temperate humid and semi-humid continental monsoon climate zone, with an annual average temperature of approximately 1.4–4.3 °C and annual average precipitation of 450–650 mm [32]. According to meteorological observation data in 2017, the meteorological elements in northern Sanjiang Plain were stable with no extreme abnormal phenomena. From the perspective

of administrative divisions, the region consists of 12 counties/cities, including Fujin, Suibin, Youyou, Huachuan, Jixian, Luobei, Hegang, Jiamusi, and Tangyuan. The cropland area of this study area was $3.14 \times 10^4$ km$^2$, accounting for 59.1% of the total land area. In terms of crop patterns, paddy, corn, and soybean were the main crops. Rice cultivation in the north of the Sanjiang Plain conforms to rice cultivation rules in cold regions [33–35], and rice is transplanted in mid May, usually between 15 May and 25 May. The harvesting date of rice was usually after 25 September and before 16 October (Figure 1).

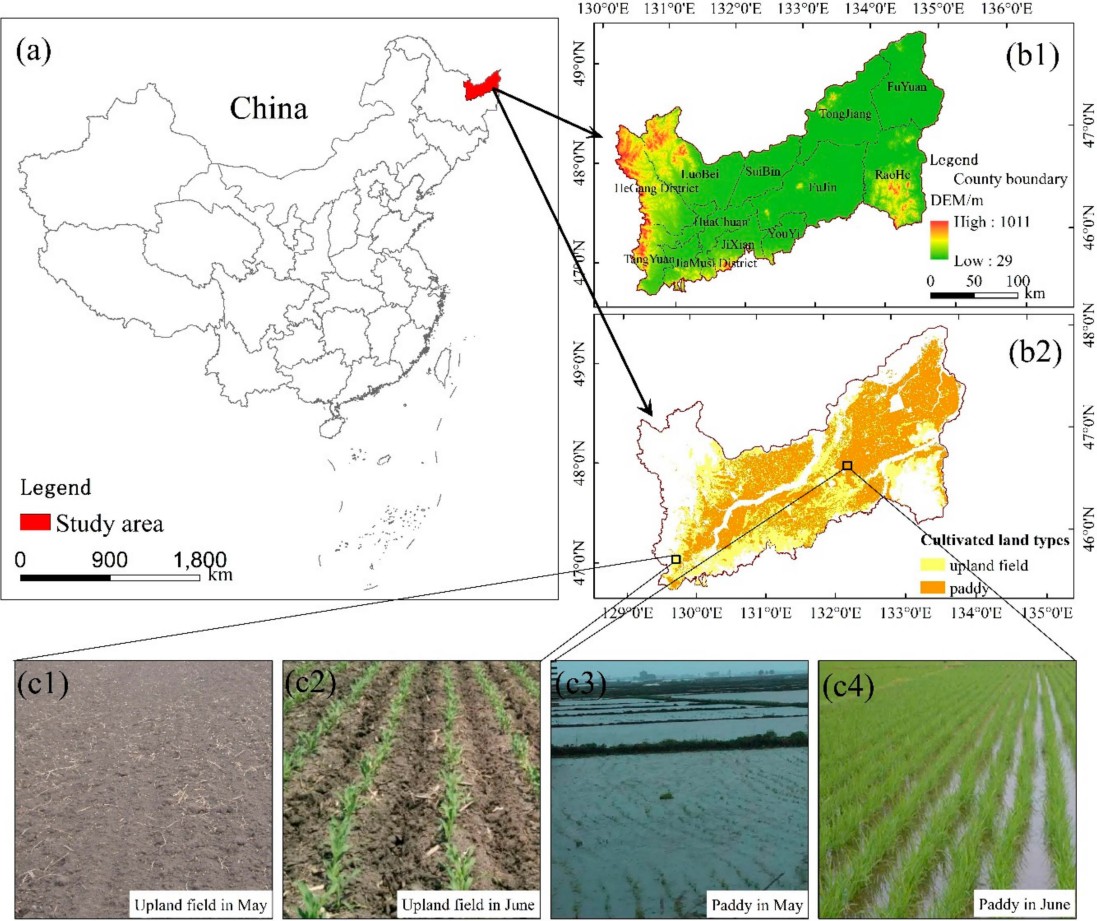

**Figure 1.** (**a**) Spatial position of the study area in China; (**b**) geographical information and cropland information, namely, (**b1**) elevation of the study area and (**b2**) cropland structure; (**c1**) and (**c2**) upland crops in early May and June 2017, respectively; and (**c3**) and (**c4**): rice paddies in early May and June 2017, respectively.

### 2.2. Data and Processing

Cropland data used in this study was based on 2017 Heilongjiang crop classification raster data. Data was obtained from Landsat OLI sensor images, and was analyzed by the method of unsupervised classification. Data analysis and information extraction was achieved using ERDAS software. Spatial resolution of the grid data was 30 m, and the overall accuracy of the sample point test was over 95%. This data accounted for 84.4% of the total cropland areas, in which three main crops were planted: rice in paddies, corn, and soybeans. Images obtained by MODIS have been widely used in LST studies [36–38]. MODIS LST day data are acquired once per day, and MOD11A2 is the maximum product synthesized from the data every 8 days. In the study, MOD11A2 data of the MODIS/Terra daytime land surface temperature from May to October in 2017 were provided by the LAADS DAAC platform [39]. For preprocessing of the MODIS data, our study re-projected data acquired from the

LAADS DAAC platform using the MODIS Reprojection Tool (MRT). Then, the merge and extraction tools in ArcGIS software were used for splicing and cropping to obtain the monthly mean LST.

*2.3. Data and Processing*

2.3.1. Establishment of Grid Units and Statistics of Paddy Grades

A reasonable setting unit is the basis for exploring spatial features and correlation analysis of factors, especially when research requires coherent analysis across different data sources [40–42]. To achieve analysis of two kinds of different data (the MODIS and Landsat data), this study used a 1 km grid unit as the basic unit of data analysis, combined with the ArcGIS software platform. Using this unit, the LST information of MODIS and land use information of Landsat were analyzed individually and comprehensively by mathematical statistical analysis. To ensure the representativeness of the unit temperature value for the LST of the cropland, the units whose water and built-up area accounted for more than 60% or whose cropland area accounted for less than 40% were excluded. Then, the effect of other land-use types on the average LST within the unit grid was excluded. For visualizing and quantifying the paddy area differences between different units, the ratio of the paddy field area to the total area of cropland was divided into 1–10 grades at 10% intervals. The paddy ratio and average LST of each grid unit was counted by regional statistical analysis. Then, further analysis of paddy distributions and the calculation of mean LST according to different paddy ratio grade scales were undertaken.

2.3.2. Correlation and Regression Analysis Method

The relationship between LST and rice paddy area was analyzed by correlation coefficients of a linear regression and by different fitting formulas. In this study, only one argument and one dependent variable were included; thus, we discussed one-way linear and one-way nonlinear regression analyses, respectively. We mainly use three regression methods (linear, quadratic, and cubic) to fit the relationship between independent variables and dependent variables, and one-way linear regression analysis mainly used the calculation of correlation coefficients and the results of significance test. The study used the correlation and regression analysis tools in SPSS to explore the correlation between paddy grades and the monthly mean LST.

2.3.3. Temperature Difference Rate Calculation

The changing rate of LST difference can quantitatively describe the cooling effects of different scales of paddy fields. The time difference of the paddy field cooling effect can be obtained by comparing and analyzing the LST difference of different months. By analyzing changes in LST differences, it can be determined if LST change time node or paddy grade node exists. The framework for analyzing different cropping patterns and their effects on LST is provided in Figure 2.

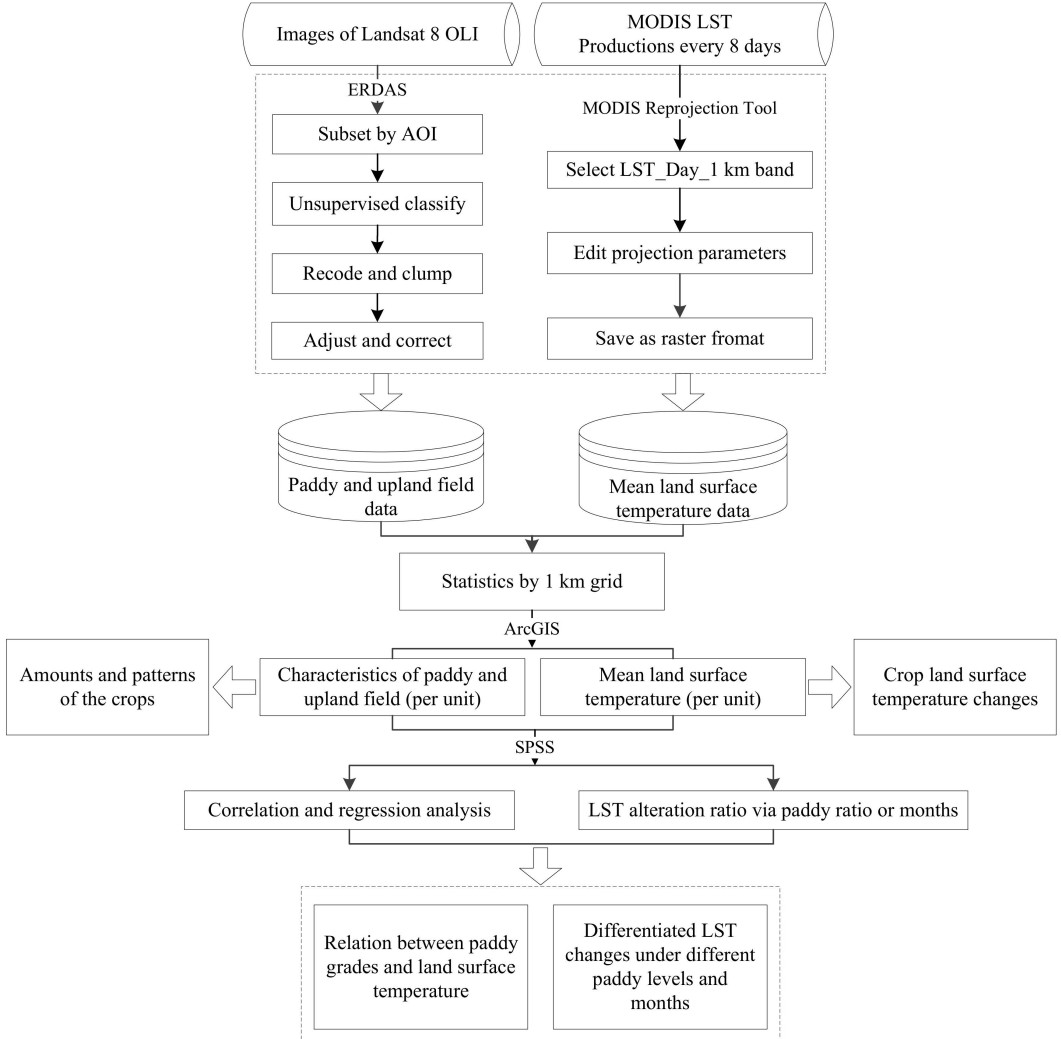

**Figure 2.** Flow chart of research steps and obtained results.

## 3. Results

### 3.1. Amounts and Patterns of the Crops in the Study Area

The northern Sanjiang Plain is a region characterized by large-scale cropland planting, in which paddy rice is the dominant crop. According to statistics for the year 2017, paddy fields covered an area of $1.77 \times 10^4$ km$^2$ and accounted for 66.8% of the total cropland area. In contrast, upland crops only covered an area of $8.81 \times 10^3$ km$^2$ and accounted for 33.2% of the total cropland area. For the crop patterns (Figure 3a), paddy fields were mainly distributed in the flat central and eastern plains, along with the aggregation patterns, while upland crops were distributed in uneven terrain regions and presented a scattered spatial distribution. Such cropland distribution characteristics in the study area were mainly caused by crop planting requirements. Research has shown that terrain has had a significant impact on the spatial distribution of paddy fields, and flat terrain has been conducive to the growth of rice [43]. Thus, the ground slope is required to be no more than 6 degrees in local planting habits, so that paddy fields are mainly distributed in flat areas. Comparatively speaking, upland regions can develop areas with higher slopes.

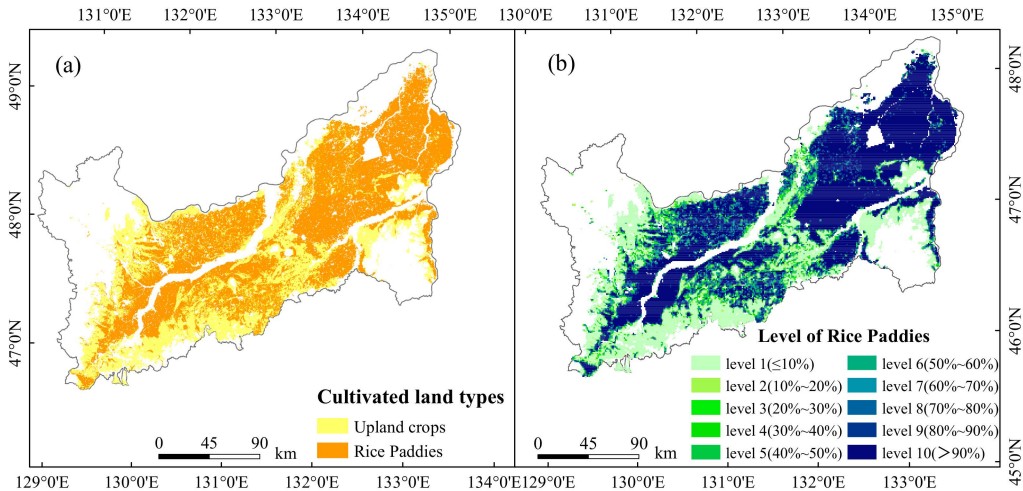

**Figure 3.** (**a**) Crop patterns and (**b**) paddy grade distributions in the study area.

The grades of paddy field areas that occupied the 1-km grid are summarized in Table 1. Their spatial distribution is displayed in Figure 3b. According to the Table 1, paddy grades were mainly distributed in levels 1 and 10, with areas of $1.33 \times 10^4$ km$^2$ and $5.29 \times 10^3$ km$^2$, respectively. These areas accounted for over 70% of total cropland areas; specifically, levels 1 and 10 accounted for 50.2% and 20.0% of the total cropland areas, respectively. In comparison, paddy fields were rarely distributed in the 2–9 grades, with graded proportions ranging from 2.6% to 7.2%. From Figure 3b, the grades of crop planting structure represented the agglomerated paddy field, the agglomerated upland crops, and the mixed paddy field and upland crops, while the paddy aggregation pattern had its advantages.

**Table 1.** Paddy grade based on the proportion of paddy area in the 1 km grid unit.

| Paddy Area Grade | Number of Cells | Agricultural Area ($10^2$ km$^2$) | Paddy Field Area ($10^2$ km$^2$) | Area Ratio Scale (%) |
|---|---|---|---|---|
| 1 | 8541 | 52.86 | 0.61 | 0–10 |
| 2 | 1236 | 8.63 | 1.28 | 10–20 |
| 3 | 989 | 7.05 | 1.75 | 20–30 |
| 4 | 974 | 6.85 | 2.40 | 30–40 |
| 5 | 1012 | 7.25 | 3.27 | 40–50 |
| 6 | 1098 | 7.90 | 4.35 | 50–60 |
| 7 | 1345 | 9.40 | 6.13 | 60–70 |
| 8 | 1718 | 12.33 | 9.30 | 70–80 |
| 9 | 2584 | 19.10 | 16.34 | 80–90 |
| 10 | 16,681 | 133.47 | 131.33 | 90–100 |

*3.2. Crop Land Surface Temperature Changes in the Growing Seasons*

The growing season of crops is from May to October in the cold regions of China, and the quantitative and spatial characteristics of the monthly average LST in crops over the growing season in the northern Sanjiang Plain are provided in Figures 4 and 5. May is the beginning of local crop phenology. For paddy fields, water is the main underlying land surface at this stage. The heat capacity of water is larger than that of other crops. Therefore, the evaporation process of paddy fields has a cooling effect on the land surface. Upland crops in May are in the stage of sowing and germination. The underlying land surface is characterized by a large area of bare soil and a smaller area of crop coverage. The LST of upland crops is higher than that of paddy fields (Figure 5a). In June, as paddy fields move to the tillering stage, paddy water still occupies a large area. Thus, the LST of paddy fields is still much colder than that of other crops, such as upland crops (Figure 5b). In July and August (Figure 5c,d), solar radiation energy increases, and the near-surface air temperature is higher than in

May and June. Changes in these factors lead to the rise of the paddy surface temperature. Meanwhile, transpiration is greatly increased in upland crops due to the improvement of vegetation coverage in the peak period of crop growth, which leads to reduced LST. However, the LST of paddy fields is still observed to be lower than that of upland crops. In September and October (Figure 5e,f), crops reach maturity and are harvested. During this period, crop vegetation coverage decreases or disappears, and the land surface temperature is mainly determined by soil moisture. Higher soil moisture occurs in paddy fields, and, therefore, the LST of paddy fields continues to show a lower trend than that of upland crops.

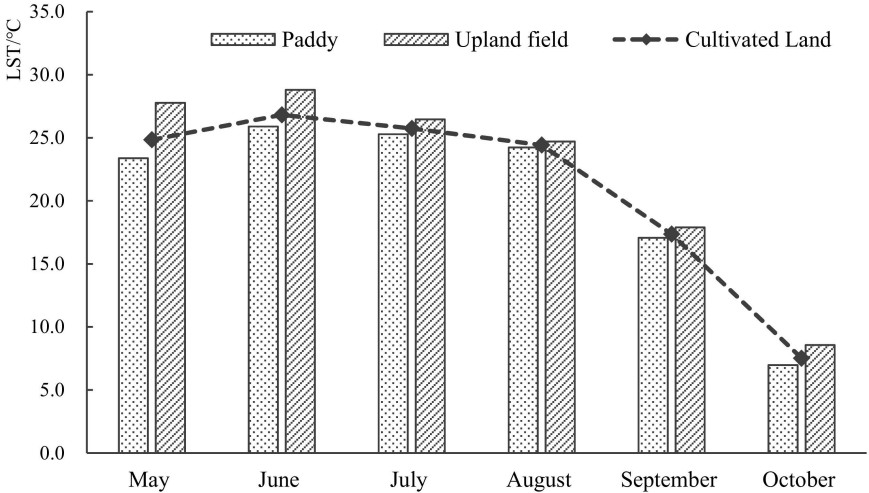

**Figure 4.** The quantitative characteristics of the monthly average land surface temperature (LST) in crops during the growing seasons.

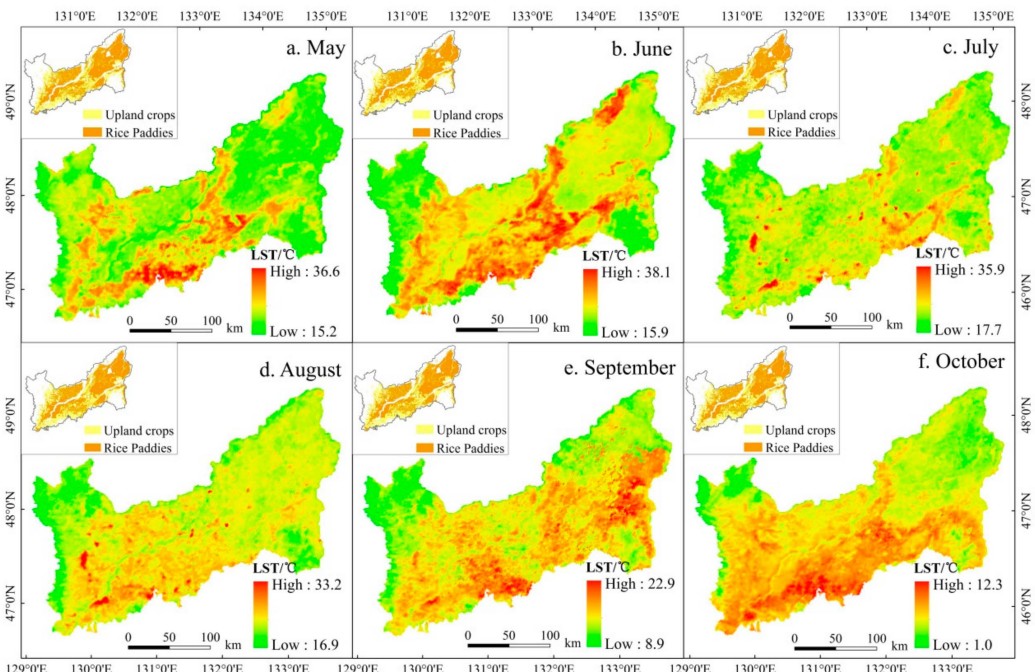

**Figure 5.** The spatial patterns of monthly average LST in crops in the growing seasons.

### 3.3. Relationship between Paddy Grades and Land Surface Temperature

Correlations of paddy grades and monthly LST are provided in Table 2. Pearson's coefficient results indicate the negative correlation between mean land surface temperature and paddy grades by month (Table 2), with significance at the 0.01 level. From the value of correlation coefficients, although there were differences among different months, the relevant coefficient values were all over 0.8. This means that the paddy had a significant cooling impact on the regional LST. From May to October, when the proportion of paddy area to the total cropland area increases by 10%, the mean LST exhibited a decreasing trend (Table 2).

**Table 2.** Correlations of paddy grades and monthly land surface temperature.

| Month | Positive(+) or Negative(−) | Correlation Factor ($r$) | Pearson Coefficient ($p$) |
|---|---|---|---|
| May | | 0.988 | <0.01 |
| June | | 0.971 | <0.01 |
| July | | 0.976 | <0.01 |
| August | | 0.852 | <0.01 |
| September | | 0.986 | <0.01 |
| October | | 0.938 | <0.01 |

Furthermore, the functional relationships between paddy grade and monthly mean LST were established using the regression analysis method. By establishing the linear and various non-linear regression models between the two factors, the functions of these two factors in different months were obtained (Table 3). Of these, the three functional relationships indicated the best correlation, namely, the linear relationship, the quadratic relationship, and the cubic relationship.

**Table 3.** Correlation coefficient ($R^2$) of best fitting degree functions in different months.

| Month | Linear | Curve | |
|---|---|---|---|
| | | Quadratic | Cubic |
| May | 0.976 * | 0.996 * | 0.998 * |
| June | 0.942 * | 0.994 * | 0.994 * |
| July | 0.952 * | 0.989 * | 0.994 * |
| August | 0.726 * | 0.861 * | 0.979 * |
| September | 0.973 * | 0.991 * | 0.992 * |
| October | 0.88 * | 0.974 * | 0.994 * |

Notes: the symbol * indicates the significance characterizing the regression model was less than 0.01.

The cubic function obtained the highest degree for all months, followed by the quadratic values. In comparison, the linear fitting indicated a significant monthly change; most of its simulation was good during the growing seasons, while in August and October the linear fitting values were lower than for the other two functions. A negative correlation between paddy grades and monthly mean LST is shown in Table 3, and the functional relationship indicated that neither was a simple linear correlation but a curve relationship. With increases of paddy grade, mean LST changes were non-uniform. The visual results of the three different fitting methods are provided in Figure 6. The curve also indicated a phenomenon, namely, as the paddy grade increased, the cooling effect of the paddy was also enhanced.

### 3.4. Analysis of Differentiated LST Changes under Different Paddy Levels

The LST in different paddy grades indicated similar trends in different months, while the LST changes had different values. The widest range of monthly mean LSTs among different paddy grades was 22.7–28.1 °C, which appeared in May, and the smallest range appeared in August, of 24.1–24.8 °C. Further quantification of the differences of paddy levels on land surface temperature could be accounted from Figure 6, consequence shows the average LST rate of change in May for the maximum (0.6 °C per

grade), followed by 0.4 °C during the growing season. This means the increases of paddy field rates in May had the most significant effect on LST. The change rate in August was observed to be the lowest, indicating the effect of paddy fields on LST was weak.

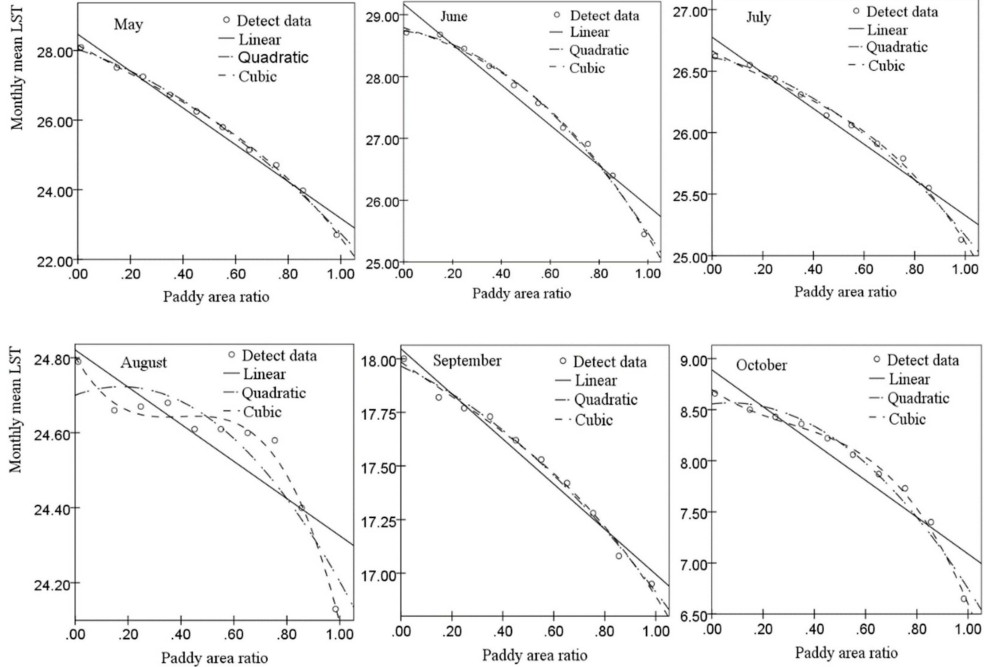

**Figure 6.** Regression results of paddy grades and the monthly mean land surface temperature in different months.

The proportion of paddy areas to total cropland over 80% in each unit was as set as the high level, and the proportion of less than 20% was set as the low level. Then, the relationship between LST in high and low levels and crops in different months was examined (Figure 7). The trend of paddy fields firstly decreased and then increased during the study period. May showed the maximum difference between the high and low levels, with LST changes of 5.1 °C, followed by June, of 3.1 °C; other months ranged from 0.6 to 1.9 °C. Overall, LST differences were evident during the growing season, and especially in May and June.

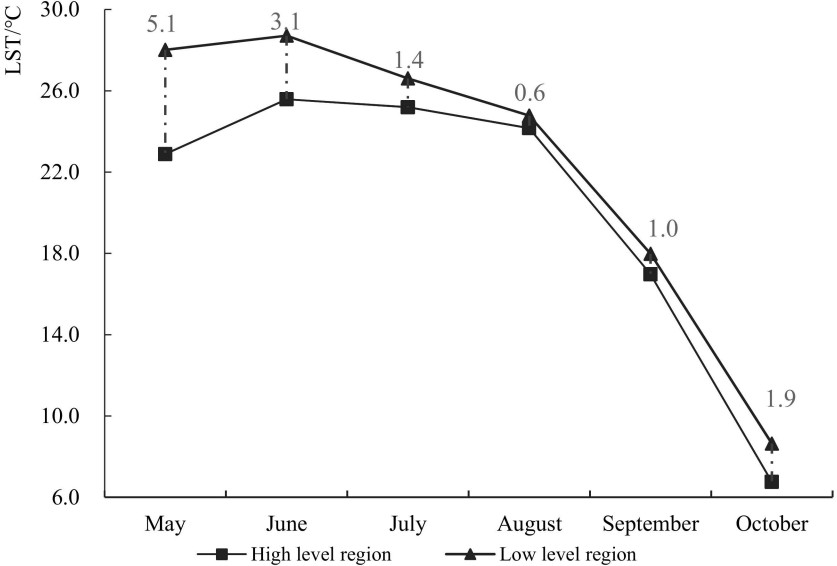

**Figure 7.** Land surface temperature in high and low level regions of paddy ratios.

## 4. Discussion

### 4.1. Cooling Effect of Paddy Fields on Land Surface Temperature and Its Significance

LST is one of important components in the energy balance of a land surface. In the process of energy flow, it evidently affects sensible heat flux and plays an important role in surface energy exchanges [44–47]. It is a significant influencing factor of the near-surface temperature and local microclimate [48,49]. Therefore, different human land uses affect the LST by changing natural land surface types. This will feedback to the local microclimate. According to the study results derived from the resolution of remotely sensed images, the monthly mean LST of higher paddy grades was lower than that of other crops during the growth season, which implies a cooling effect of paddy expansion.

Research on LST characteristics and its influencing factors in cropland has practical significance and theoretical guidance for regional agricultural production. The growth period of crops in China's cold region is mainly focused in the months of May to October. Therefore, the key processes, including rice tillering, maize stem, and leaf growth, generally take place from May to August each year. In the growing season, crop growth has a great impact on the total production [50,51]. During this period, temperature was the main restrictive factor for crop growth, especially in the cold region. Due to the limitation of hydrothermal conditions, crops in cold regions are more sensitive to temperature differences during their growth [52,53]. Thus, the cooling effect of large scales and agglomerations of paddy fields will feedback to the growth of the crops. The meteorologically adverse phenomenon caused by human activities and its effect on agricultural production cannot be ignored.

### 4.2. Threshold of the Effect of Paddy Field on Land Surface Temperature

The scale and time thresholds were found in the results of LST change trends on the paddy grade scale from the study. As the results show, the changes of land surface temperature were more significant with the increases of paddy grades. In addition, if the paddy grade exceeded level 8, the average land surface temperature decreased with a faster rate, which exceeded the monthly average rate. Thus, the scale threshold (level 8) of the land surface temperature led to a fast cooling effect. Monthly LST values decreased more obviously after the paddy field area ratio exceeded the threshold. From the perspective of crop growing seasons, May and June were the two obvious months in which the cooling effect of the paddy field was characterized by an average temperature range of 5.1 °C (Figure 7). Considering the impact of LST on crop growth and total grain production, the paddy planting scale must be set reasonably in the process of optimizing the quantity and spatial pattern of different planting types. Therefore, the large-scale expansion of paddy fields should be taken as one of the indicators in the planning of cropland planting. Exploration of the rules and features of paddy expansion effects on land surface temperature can provide further theoretical guidance for cropland planning.

### 4.3. Research Limitations and Future Prospects of the Paddy Expansion

Although the cold effect of paddy expansion on LST has been captured in this research, subsequent studies are required. The time period of this study focused on the growth period of crops from May to October. Although no extreme weather conditions occurred during the study period, multiple annual results of paddy expansion still need to be further discussed. A 10% standard paddy area ratio grade was found, but whether different spatial scales are suitable still needs to be further analyzed. In addition, the soil property and other processes that may affect the radiation balance need to be further discussed. The influencing factors involved in the process of surface energy exchange are complex, and surface temperature, wind speed, terrain, and other factors should be taken into account in follow-up research [54–56]. In subsequent research on surface energy exchange, attention should be paid to the comprehensive processes of various factors, including LST, as well as to advanced mathematical models [57,58]. Furthermore, the factors affecting crop yield in the study area are relatively complex.

In future studies, the controlling variable approach should be used in the investigation of effects of paddy expansion on crop growth and grain yield security.

## 5. Conclusions

The present study investigated the relationship between LST and paddy expansion, derived from remotely sensed products of Landsat OLI and MODIS LST. The following results were obtained: for cropping patterns, cropland planting was given priority to paddy fields, accompanied by the agglomeration and expansion patterns, while upland crops presented discrete patterns. For LST changes in the growing season, the maximum LST occurred in June, and the lowest values occurred in October in the whole region. The LST of paddy fields was observed to be lower than that of upland crops in the whole growing season. Furthermore, at the 1-km grid scale, the monthly LST values and the paddy field ratios displayed a significant negative relationship, and the relationships obtained were better represented by a cubic function than linear fitting. Finally, the threshold value of the paddy expansion effect on fast cooling of land surface temperature was level 8 in the northern Sanjiang Plain. The findings of this paper are of great significance for guiding agricultural production and studying the environmental effects of paddy expansion in cold regions.

**Author Contributions:** All authors have contributed to the intellectual content of this paper. Conceptualization, Validation and methodology, G.D.; Writing—original draft preparation and data curation, W.L.; Writing—review and supervision, T.P.; Software and visualization, H.Y.; Data curation, Q.W.

**Funding:** This research and the APC were funded by Chinese National Science Foundation, grant number 41571167.

**Acknowledgments:** The authors would like to thank the anonymous reviewers and handling editors for their constructive comments, which greatly improved this article from its original form.

**Conflicts of Interest:** The authors declare no conflicts of interest.

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
