# Peer review of "Cooling Effect of Paddy on Land Surface Temperature in Cold China Based on MODIS Data: A Case Study in Northern Sanjiang Plain"

_sustainability, doi:10.3390/su11205672_

Round 1

Reviewer 1 Report

Thanks very much authors for this excellent research paper.  I would like to recommend a few changes to enhance the quality of the manuscript.

Only a paltry number of references have been included in the text, which warrants a thorough literature review and inclusion of relevant bibliography. Devise a flowchart to delineate the methodology deployed in this research work explicitly. Include a clearly defined methodology section.  Some of the keywords exactly match with the title of the paper. Either you change the title or the keywords. None of the images (Figures 1- 6) are crisp and look extremely hazy. They are not legible.  Please provide figures in high resolution.  Data representation and visualizations have been scanty in the manuscript. Provide more graphs supporting the data of the tables. Revise the entire manuscript thoroughly to improve the English Language. Propose a coherent model supported by graphics for your research work before the conclusion part deploying present MODIS and Landsat data.

Author Response

Responses to Reviewer #1:

Our team is very grateful to reviewer#1 for his (or her) responsible work. The comments improved the quality of the paper and helped us. According to all the comments given by reviewer#1, we realized that there were many important problems in our paper, including insufficient references, insufficient graphs, insufficient details, and grammatical errors. According to the reviewers' comments, our responses are as follows.

Comment 1: Only a paltry number of references have been included in the text, which warrants a thorough literature review and inclusion of relevant bibliography.

Response: Thank you for your suggestion. We added references and revised comments in introduction, materials and methods, and discussion.

In introduction, we added references 3-5 (L41-L44, page 1), 8-9 (L46-L47, page 2), 11-21 (L52-L55, page 2), 22-23 (L64-L65, page 2), 28-29 (L70-L72, page 2);

In materials and methods, we added references 34-35 (L96-L97, page 3), 37 (L112-L113, page 3); 41-42(L122-L123, page 4); 43(L170-L172, page 5);

In discussion, we added references 45-47 (L273-275, page 11), 54-56 (L313-315, page 12), 57-58(L315-317, page 12).

Comment 2: Devise a flowchart to delineate the methodology deployed in this research work explicitly. Include a clearly defined methodology section.

Response: Thanks for your valuable comments. Based on the work flow of our whole research, we designed a flow chart (figure 2) and inserted it into the manuscript (L159, page 5). In this flow chart, we reflected the research steps in sequence and marked the results obtained by each step, so that all the processes in the research could be visual and more clearly understood.

Comment 3: Some of the keywords exactly match with the title of the paper. Either you change the title or the keywords.

Response: We are very grateful to you for your valuable advice, and modified the keywords (L33-L34, page 1).

Comment 4: None of the images (Figures 1-9) are crisp and look extremely hazy. They are not legible. Please provide figures in high resolution.

Response: Thank you very much for your kind comment. We modified all the images again, and improve their qualities. Now all the images in this manuscript had been updated with higher resolution of 450 dpi (Figures 1-9).

Comment 5: Data representation and visualizations have been scanty in the manuscript. Provide more graphs supporting the data of the tables.

Response: Thank you for the comment. We visually processed the necessary table data in the manuscript and presented some table data in the form of graphs, and the main diagrams are added by figure 4 (L187, page 6), figure 8 (L259, page 10), and figure 5 (L212, page 7). The corresponding contents are respectively the proportion of cultivated land and paddy field area corresponding to different paddy grades in table 1, the LST changing ratio of different paddy levels in different months in table 4, and the average land surface temperature of paddy and upland in different months.

Comment 6: Revise the entire manuscript thoroughly to improve the English Language.

Response: Thank you for the comment. We turned to MDPI's professional language service to improve the language level of the manuscript. English-editing-certificate (Please see the attachment).

Comment 7: Propose a coherent model supported by graphics for your research work before the conclusion part deploying present MODIS and Landsat data.

Response: Thank you for your comments. In order to achieve coherently analysis of MODIS and Landsat data, we built 1km grid unit and took this grid as the basic unit of data analysis combining with the GIS software platform. Based on this unit, LST information of MODIS and land use information of Landsat were analyzed by mathematical statistical approach. In this way, we realized the collaborative processing of data from different data sources. According to your comments, we found that the description of the method and data processing process was not detailed enough, thus, we further refined the method part (L123-127, page 4). At the same time, we presented it in new figure 2 (L159, page 5).

Reviewer 2 Report

See attached file

Author Response

Responses to Reviewer #2:

Our team is very grateful to reviewer#2 for his (or her) responsible work. The comments improved the quality of the paper and helped us. According to the comments of reviewer 2, we found that there were many problems in details in our paper. According to the reviewers' comments, our responses are as follows.

Comment 1: Both English and the resolution of figures need to be improved considerably.

Response: Thank you for your comments. We modified all the images and improved their qualities. Now all the images in this manuscript had been updated with higher resolution of 450 dpi (Figures 1-9). As for the English language, we turned to MDPI's professional language service to improve the language level of the paper. English-editing-certificate (Please see the attachment).

Comment 2: One factor that the authors should pay attention is that LST is an important factor to determine sensible heat flux, but heat exchange (mainly determined by wind speed and surface roughness) is also an important factor to determine sensible heat flux. If heat exchange does not occur (think about calm and cool night) then you may observe cool surface and warm air temperature. For example, Ikawa et al. (2018) reports difference in surface temperature depends on wind speed. The higher wind speeds, the more differences in surface temperature between two different surfaces.

Response: We added the content at L313-L317 (page 12), and appreciated for your constructive comments. This is a good point that should not be ignored. The influencing factors involved in the process of surface energy exchange are complex, and surface temperature, wind speed, vegetation and other factors should be taken into account in the follow-up research. In the subsequent further research on the surface sensible heat flux change and the surface energy exchange, attention should be paid to the comprehensive action process of various factors including LST.

Comment 3(L48-L52): There may be more studies on GHG of rice paddy fields, but there are also many studies about impacts of crop fields on air temperature (Ikawa et al., 2018; Nocco et al., 2019) and those are more appropriate to review here than studies on GHG.

Response: Thank you so much for your comments. The contribution process of paddy fields to GHG is an important manifestation of paddy fields’ climate effect, which will have an important impact on the surface thermal environment. Meanwhile, the impact of crop fields on air temperature and humidity is also closely related to this study. Therefore, on the basis of summarizing the research on paddy fields’ GHG emission, we supplemented the relevant references of the research on crop fields’ influence on air temperature, humidity and other climatic factors. (L50- L55, page 2)

Comment 4(L58): here and elsewhere (look at reference too). Please capitalize letters when necessary!

Response: We carefully checked all the words in the text and capitalized where necessary, including the word “Sanjiang” in L21 and L64 (page 1-2).

Comment 5(L71): different levels? What do you mean by it?

Response: We are sorry for the unclear narrative here. In order to visualizing and quantifying the paddy area differences between different units, we divided paddy field area ratio into 1~10 grades at 10% intervals, which characterized by 1-10 levels. We explained in L131-L133 (page 4). We exchanged “different levels” to “different paddy field ratios” (L81, page 2).

Comment 6(L86): What are typical rice transplanting and harvesting dates?

Response: Thank you for your remind about this. The rice cultivation in the north of Sanjiang Plain conforms to the rice cultivation rules in cold region, and rice transplanted in mid-May, usually between May 15 and May 25. The harvesting date of rice usually after September 25 and before October 16. We added the content in the manuscript (L96-L99, page 3).

Comment 7(L93): obtained by ERDAS? ERDAS is a software to analyze image.

Response: Thanks for your comments. The existing description tends to confuse the concepts, so we updated the description of relevant content in L108-L110 (page 3). The cropland data was obtained by the Landsat OLI sensor images, and the method of data information extraction was unsupervised classification. The analyze tool was ERDAS software.

Comment 8(L96): 85.59% may be too precise. Here and elsewhere please consider significant digits. Please remember a very good temperature has an accuracy of about 0.1 dC.

Response: We accepted the suggestions about the significant digits, and updated the accuracy of percentage data to tenths’ digit, mainly distributed in L95 (page 3), L111 (page 3), L165 (page 3), L166 (page 3), L181 (page 6). Meanwhile, the accuracy of all temperature data was adjusted to 0.1 dC, and the updated data were mainly distributed in L252 (page 9), L253 (page 9), L255 (page 10), L265 (page 10), L266 (page 10).

Comment 9(L97): Remove the explanation of LST. It does not convey any meaning

Response: Thank you for your comments. We have deleted the explanation of LST at L112 (page 3).

Comment 10(L101): high temporal resolution. So what was the temporal resolution? How often were data retrieved each day?

Response: Thank you for this comment. We supplemented the manuscript with information about the temporal resolution of the data (L113-114, page 3). MODIS land surface temperature day data is acquired once a day, and MOD11A2 is the maximum product synthesized from the data every 8 days. According to this comment, we realized that the description of the research data was not detailed enough, which might lead to the loss of some important data presentation information.

Comment 11(L122-133): Delete all. Everyone can learn what correlation analysis is from Wikipedia etc.

Response: Thank you for your comment. We deleted all the content that was pointed out, and updated on the use of correlation analysis in this article (L138-L142, page 4).

Comment 12(L157): here and elsewhere. What is hm? Please use km instead.

Response: Thank you for your comment. We scanned all over the manuscript and converted all the hm2 covered in the article into km2 (L68, page 2; L95, page 3; L165, L166, page 5; L179, L186, page 6 ).

Comment 13(L160): Check grammar. While…

Response: Thanks for your kind reminder. We carefully corrected the grammar, and rephrased (L198, page 8).

Comment 14(L163): regulation by what?

Response: What we wanted to express in this part was the planting habits of local farmers.It always included restrictions about local rice cultivation, based on the need suited to characteristics of paddy, such as the requirements of the land slope and so on. Since the terrain was significantly influenced the rice growth, local planting habit of the slope is no more than 6 degrees, cultivated distribution in study area exhibited that paddy was in the flat central and eastern plains along with the aggregation, while upland crops distributed in uneven terrain regions and presented a scattered spatial distribution. The sentence here is not clear enough which may lead to confusion, so we rephrased in L169-L174 (page 5).

Comment 15(L180): Were rice fields always flooded? Do farmers implement drainage in the middle of growing seasons to control growth?

Response: Thanks for your comment. The statements here are not clear enough that caused confusion. We have rephrased in L193 (page 7). To control rice growth, local farmers do implement drainage in the middle of growing seasons (usually in late June). May is the period in which local farmer flooded paddy fields and transplanted rice. So the water is the main underlying land surface of paddy field at this period.

Comment 16(L183): You do not say bare soil if plants were grown.

Response: Thanks for your comment. The description here was not clear enough, which led to the misunderstanding of the review. We modified the sentence and explained it in more detail (L197-198, page 7). Under the influence of climate, dryland crops in the study area usually begin to be sown in May, when the crops were in the stage of seed germination, and seedlings appeared in the late may. At this time, the crop canopy area was small, and the exposed soil dominated the whole underlying surface. Therefore, during May, the upland underlying surface was characterized by a large area of bare soil and a smaller area of crop coverage.

Comment 17(L187): LST became higher in July and August because air temperature was also higher? Do you think that the smaller difference is also due to other crops reaching? Please remember that at the peak growing season, transpiration dominates in evapotranspiration (Wei et al., 2015). Ikawa et al. (2017) shows that evapotranspiration from a rice field increased as LAI increased.

Response: Thank you for your very instructive suggestions. We updated the relevant parts of the article in L201-L206 (page 7). In view of this opinion, we have consulted a lot of relevant articles to explore and analyze the surface temperature changes in July and August and the reasons for the narrowing of the temperature difference. In July and August, solar radiation energy increases. Due to the simultaneous decrease of surface water in paddy fields, the energy loss caused by evaporation process is relatively reduced. The cooling effect of water evaporation is greater than that of transpiration of crop leaf area, which leads to the rise of paddy surface temperature. However, transpiration is greatly increased in upland due to the improvement of vegetation coverage in the peak period of crop growth, which leads to the phenomenon that the surface temperature is lower than that in May and June. Thanks again for your suggestions.

Comment 18(L250): Here and elsewhere. You do not need .  between Table and numbers.

Response: Thank you very much for the kind reminding. We carefully checked all the “Table” and “Figure” mentioned in manuscript, and delete all the . (period) that were unnecessary (L198, L200, L207, page 7; L220, L224, page 8).

Comment 19(L296): including.. did you use anything else?

Response: Thank you for your kind reminding. We deleted the word “including” in L323 (page 12) for there were only two types of remotely sensed products.

Comment 20: References: Again please make sure to capitalize when it is necessary.

Response: We have updated all the reference formats according to the MDPI standard and make sure to capitalize when it is necessary. Thanks again for your kind comment.

Round 2

Reviewer 1 Report

Dear authors, I am okay with the changes made. Many thanks.

Author Response

Our team is very grateful to reviewer#1 for his (or her) responsible work. His (or her) support and affirmation of this manuscript have greatly improved the quality of our article and provided an effective reference for our future research and paper writing.

Reviewer 2 Report

see attached file

Author Response

Response to Reviewer 2 Comments

Our team is very grateful to reviewer#2 for his (or her) valuable help. And his (or her) responsible attitude and respect for scientific research moved us. The comments improved the quality of the paper. According to the comments of reviewer 2 in round 2, we found that there were still many problems in details in our paper, and our responses are as follows.

Point 1: One major point related to its content I would like to point out is in L261 the authors argued paddy surface temperature increased because of less water surface. But are you able to rule out this is simply because air temperature (or solar radiation) also increased? When rice canopy is developed with LAI of greater than around 3 or 4, then daily maximum transpiration can be even greater than evaporation from water surface, though water surface would evaporate more when averaged daily.

Response 1: According to your opinion, we have rewrote this content (L214-218, page 7), and explained the main reason for the increase of the surface temperature of the paddy field during this period as the increase of the air temperature and solar radiation energy.

Point 2: Also I have a feeling that there are too many figures or tables for the short paper, and some of them are quite redundant and you may want to delete or forward to supplemental materials. Fig. 4 is totally unnecessary given Table 1. The authors evaluated how much LST changes by the change in paddy area fraction in Table 4 and Figure 8. I think that Table 4 and Figure 8 are not necessary if the range of y axis on Figure 7 is fixed and the authors discuss based on the slope of the regression lines. For example, I would suggest 22 – 29 deg for May and August, 24 – 31 deg for June and so on. Please remember that many of us evaluate the quality of manuscript based on the simplicity of research paper. Please minimize the volume of the paper as much as possible.

Response 2: In order to make the article more concise, we have deleted Figure 4 (L200, page 7), Figure 8 (L273, page 11) and Table 4 (L272, page 11), also have updated the figure and table names in the manuscript. Thank you for your advice. We will also adopt your advice in future manuscript writing.

Point 3 (L17):. have been evolved

Response 3: Thank you. We have changed the grammar in L17 (page 1).

Point 4 (L35): abstract needs to be standalone – this means that readers should not have to read the main text and understand abstract. You should not use Level 8 without explanation here. Instead what about rewriting something like this, Finally, LST decreased with the increased fraction of rice paddy area more rapidly when rice paddy was aggregated and accounted for more than 80% of each study grid.

Response 4: We have updated the sentence follow this very appropriate advice in L30-32 (page 1). Thank you.

Point 5 (L57): Please let me rewrite a bulk of sentences appeared hereafter: A number of studies have investigated the effects of croplands on surrounding environment. For example, irrigated agriculture moderated daily maximum air temperature and increased daily minimum air temperature, narrowing the range of diurnal change in air temperature [11]. The extent of the effects of rice fields on air temperature and humidity can vary by growth stages and by cultivars with different physiology [12, 13].

Response 5: We are very grateful for your kind help. These sentences make the content more clear, and we updated the sentences in L52-61 (page 2).

Point 6 (L94): and elsewhere: feedback

Response 6: We have rewrote the words in L80 (page 2), L290 (page12) and L302 (page12). Thank you.

Point 7:(L170 – L190): Delete these sentences because those do not convey any further information than Wikipedia. Instead, I would suggest something like, the relationship between LST and rice paddy area was analyzed by correlation coefficients of a linear regression and by different fitting formulas. (Then you should explain you tried three different regressions)

Response 7: Your suggestion gives us a good idea to modify the expression of the research method. Follow this advice, we deleted these sentences, integrated the content of correlation analysis into regression analysis, and then rewrote the content about correlation analysis and regression analysis (L143-160, page 4). Thank you.

Point 8 (L247): Sanjiang Plain or Sanjiang plain?

Response 8: By referring to the literature, we identified this word as “Sanjiang Plain” and updated all relevant positions (L21, page 1; L70, L73, page 2; L95, L98, page 3; L175, page 6) in the manuscript. Thank you for your comments.

Point 9 (L368): Fig. 7??

Response 9: The graph corresponding to this part is figure 7 (previous figure 9) after the update of graph. We are sorry that we ignored the change here when updating the graph. (L313, page 13).

Point 10 : Fig. 2 Capitalize AOI; Mean land surface temperature or LST; Statistics

Response 10: Thank you for your advice. We have updated the words in Figure 2 (L170, page 5).

Point 11 : Table 2 Positive (+) or Negative (-)

Response 11: Thank you for the kind reminding. We have added the symbol in Table 2 (L239, page 9).

Round 3

Reviewer 2 Report

I think now the paper can be accepted.